# Cytomegalovirus Glycoprotein B Genotype in Patients with Anterior Segment Infection

**DOI:** 10.3390/ijms24076304

**Published:** 2023-03-27

**Authors:** Chu-Yen Huang, Yu-Chun Cheng, Yih-Shiou Hwang, Eugene Yu-Chuan Kang, Ching-Hsi Hsiao

**Affiliations:** 1Department of Ophthalmology, Chang Gung Memorial Hospital, Linkou Medical Center, Taoyuan 333, Taiwan; hfoagf@gmail.com (C.-Y.H.);; 2College of Medicine, Chang Gung University, Taoyuan 333, Taiwan

**Keywords:** cytomegalovirus, glycoprotein B genotype, anterior segment infection

## Abstract

(1) The glycoprotein B (gB) on the viral envelope, encoded by the most widely characterised polymorphic gene, *gpUL55*, is responsible for cytomegalovirus (CMV) entry into the host and could serve as a potential marker of pathogenicity. The aim of the present study is to investigate the distribution of the CMV gB genotype in anterior segment infection in Taiwan and its correlation with clinical manifestations and outcomes. (2) Fifty-seven patients with CMV anterior segment infection were identified according to clinical features and positivity for CMV DNA in aqueous humour samples. CMV gB genotyping was performed through polymerase chain reaction assays. Patients’ medical records were retrospectively reviewed. (3) Among the 57 aqueous humour samples tested for gB, 40 (70.28%) had multiple gB genotypes, and only 17 (29.82%) had a single gB genotype. Compared with single-genotype infection, multiple-genotype infection was correlated with higher CMV loads (*p* < 0.001) but not correlated with outcome. A higher proportion of patients with the gB3 genotype had received filtering surgery before antiviral treatment than those without the gB3 genotype (*p* = 0.046). (4) Multiple-genotype infection was highly prevalent in CMV anterior segment infection in Taiwan, and gB1 and gB3 were predominant. Multiple-genotype infection was correlated with higher CMV loads but not with specific clinical manifestations or prognostic outcomes. The gB3 genotype may be correlated with poor intraocular pressure control.

## 1. Introduction

Cytomegalovirus (CMV), a member of the herpesvirus family, is a common human pathogen that infects most of the adult population. CMV establishes lifelong latent infections in myeloid and dendritic cell progenitors after the primary infection [1]. Similar to other herpesviruses, a low level of viral gene transcription occurs in latent CMV infection [2]. CMV causes retinitis in immunocompromised patients. However, recent studies have reported an increasing number of cases of CMV anterior segment infection, including anterior uveitis and corneal endotheliitis, in immunocompetent patients [3,4]. Although CMV anterior segment infection is relatively uncommon, it can cause severe complications such as corneal decompensation and glaucomatous optic neuropathy, leading to vision loss. Diagnosis of CMV anterior segment infection is based on both clinical findings and polymerase chain reaction (PCR) confirmation of CMV DNA in the anterior chamber. No established treatment guidelines exist for CMV anterior segment infection. The current mainstay of treatment for CMV anterior segment infection is systemic ganciclovir therapy [5], but topical ganciclovir therapy is promising; however, the rate of recurrence after the cessation of anti-CMV therapy was relatively high [6,7,8]. 

The human CMV genome encodes numerous glycoproteins, including glycoprotein B (gB), glycoprotein H (gH), glycoprotein L (gL), and glycoprotein O (gO). Glycoprotein B found on the CMV envelope, encoded by the *UL55* gene, has been extensively studied, and is known to play a critical role in various viral activities such as viral entry into hosts, cell-to-cell viral transmission, and the fusion of infected cells [9]. The CMV gB gene has the potential to serve as a potent marker of pathogenicity and a viable target for treatment; vaccine-targeting gB was found to reduce post-transplantation CMV viremia in phase II clinical trials [10]. There are five primary gB genotypes, namely gB1, gB2, gB3, gB4, and gB5, identified so far [11,12]. Studies have well investigated the association between these gB genotypes and their clinical manifestations and outcomes and have attempted to determine the characteristics of specific CMV-associated conditions such as congenital CMV infection, pneumonitis, retinitis, or viremia after organ transplantation [13,14]. Nevertheless, little is known about the distribution of gB genotypes in CMV anterior segment ocular infection and its correlation with clinical manifestation; to date, only two studies have shown gB1 and gB3 predominance in CMV anterior segment infection and were both with small sample sizes [15,16].

Accordingly, the aim of this study was to investigate the gB genotype distribution in CMV anterior segment infection in the Taiwanese population and to examine the correlation between clinical characteristics and different genotypes. The study findings may elucidate the pathophysiology and offer potential new therapeutic targets for CMV anterior segment infection.

## 2. Results

In total, 57 eyes of 57 patients with CMV anterior segment infection were included in this study. All of the patients were immunocompetent. Of the patients, 12 were women and 45 were men, and their mean age was 58.7 ± 13.4 years (Table 1). Moreover, 35 (61.4%) and 22 (38.6%) patients manifested corneal endotheliitis and anterior uveitis, respectively. The coexistence of Epstein–Barr virus (EBV) DNA was detected in the aqueous samples of eight patients (14.0%). Fifteen patients (26.3%) experienced recurrence within 1 year. Furthermore, 35 patients (61.4%) required long-term antiglaucoma medications, and 10 patients (17.5%) required filtering surgery for Intraocular pressure (IOP) control. Eight patients (14.0%) had corneal decompensation and were waiting for or had already received corneal transplantation. 

### 2.1. Distribution of CMV gB Genotype

Table 2 lists the distribution of CMV gB genotypes for the 57 patients. Among the patients, 40 (70.3%) had CMV anterior segment infection involving multiple gB genotypes, and only 17 (29.8%) had infection involving a single gB genotype, namely gB1 (*n* = 4, 7%) or gB3 (*n* = 13, 22.8%). Additionally, 21 (36.8%) and 19 (33.3%) patients had infections involving two and three distinct gB genotypes, respectively. Overall, gB3 (*n* = 39) was the most common genotype, followed by gB1 (*n* = 38), gB4 (*n* = 33), and gB2 (*n* = 6). No gB5 was detected in the sample.

### 2.2. Single Genotype Infection and Multiple Genotype Infection

We compared patients with an infection involving a single gB genotype and those with an infection involving multiple gB genotypes. We observed that the patients with an infection involving multiple gB genotypes had higher viral loads than did those with an infection involving a single gB genotype (Median 56.0 [6.7, 252.0] vs. 4.6 [1.6, 8.0] copies/μL, *p* < 0.001; Figure 1). We observed no significant differences in age, sex, systemic history and immune status, ocular history (glaucoma, surgical history, or medication use), clinical features, diagnosis, coexistence with EBV, or prognosis between these two groups of patients (Table 1). Further, we also analysed the correlation between viral load (log-transformed) and prognostic outcomes using logistic regression, including recurrence within 1 year (*p* = 0.585), the need for antiglaucoma medications (*p* = 0.200), filtering surgery (*p* = 0.934), or corneal decompensation (*p* = 0.561), and we found no significant difference in all of the listed outcomes.

### 2.3. Characteristics and Prognosis with Patients Involving Different Genotypes

Because 70.28% of the patients had CMV anterior segment infection involving multiple genotypes, identifying the influence of a single genotype was complicated. Accordingly, to determine whether certain gB genotypes would correspond to certain clinical manifestations, we divided the patients into the following groups: gB1, non-gB1, gB3, non-gB3, gB4, and non-gB4 groups. We observed no significant differences between the presence of gB1, gB3, or gB4 and the manifestation of CMV corneal endotheliitis or anterior uveitis (Table 3). Most of the patients required topical antiglaucoma medications regardless of which genotypes were detected in their aqueous humour specimens. However, we found a significantly larger proportion of patients with the gB3 genotype, compared to those without the gB3 genotype, underwent filtering surgery prior to diagnosis of CMV anterior segment infection (*p* = 0.046). Specifically, all of the eight patients who received filtering surgery before antiviral treatment started were gB3 positive, which indicated poorer IOP control while antiviral treatment was not started yet. In addition, we observed no significant differences in prognostic parameters between the patients with the various genotypes. 

### 2.4. Linear Regression Model

Table 4 shows the summary results of the association between viral load (log-transformed) and multiple genotypes in the linear regression models. The results demonstrated that multiple genotypes were independently associated with viral load when adjusting for endotheliitis (regression coefficient [*B*] = 2.55, 95% confidence interval [CI] = 1.35 to 3.76), gB1 genotype and endotheliitis (*B* = 2.40, 95% CI = 0.94 to 3.87) or adjusting for gB3 genotype and endotheliitis (*B* = 2.48, 95% CI = 1.26 to 3.70). However, multiple genotypes failed to demonstrate an association with viral load when adjusting for the gB4 genotype and endotheliitis (*p* = 0.374).

## 3. Discussion

This study is the first to investigate the distribution of gB genotypes in CMV anterior segment infection in the Taiwanese population. Our findings reveal that multiple-genotype infection, correlated with higher CMV loads, was predominant (approximately 70%) in our patients. Single-genotype infection involved either gB1 or gB3. Additionally, infection involving gB3 was correlated with a higher rate of filtering surgery before antiviral treatment started, which might imply poorer IOP control without antiviral treatment; nevertheless, no other correlation was observed between clinical manifestations and different genotypes.

Our study indicated a predominance (70.28%) of CMV anterior segment infection involving multiple genotypes in the Taiwanese population, in contrast to studies conducted in other countries. Oka et al. in Japan and Zhai et al. in China have individually investigated the distribution of gB genotypes in 14 immunocompetent patients with CMV anterior segment infection each, reporting that only the gB1 (primarily) or gB3 genotype was solely present in the aqueous humour samples [15,16]. In addition to gB1 and gB3, we detected gB2 and gB4 in the samples collected from our patients; these genotypes were detected only in those with multiple-genotype CMV infection. In fact, multiple-genotype infection is not uncommon in immunocompromised patients such as those with AIDS or those who have received organ transplants with CMV infection [17,18,19,20]. Studies have identified multiple genotypes in 16.0–59.6% of patients with CMV viremia who had received organ transplants [19,21]. However, all of our patients were immunocompetent. Whether the high prevalence of CMV infection (over 90%) in Taiwan could have partially contributed to the high proportion of multiple-genotype infection in our patients is unclear. Another possibility would be the genetic diversity following primary infection. There was a higher rate of multiple-genotype infection reported in a longitudinal follow-up in urine, saliva, and plasma samples [22]. Further research of longitudinal analysis should be executed to examine this phenomenon.

In this study, the patients with multiple-genotype infection had significantly higher viral loads than those with single-genotype infection, but these two groups of patients did not differ in clinical features. Due to the sensitivity limit of the PCR, it is likely that multiple genotypes are predominantly found in samples with higher CMV load. In systemic CMV diseases, patients with an infection involving multiple gB genotypes experience CMV disease progression, higher graft rejection rates, and higher CMV loads compared with those with single-genotype infection [20,21]. Some studies reported that the amount of CMV DNA was significantly associated with the frequency of recurrence and IOP elevation and endothelial loss in CMV corneal endothiliitis [23]. Our study did not find any evidence to suggest that multiple-genotype infections lead to a worse outcome. We observed large variations in viral titres in the patients, possibly due to the patients being examined at different stages of the chronic course of the disease, which may explain the discrepancy in results between our study and previous studies in systemic disease. Additional longitudinal analyses of viral loads or peak viral loads should be performed to support the results of our study. 

The presence of gB3 was correlated with the need for filtering surgery before antiviral treatment, which indicated the importance of early diagnosis. However, we did not observe a significant difference between the gB genotypes in terms of their association with other clinical features. The Chinese study analysed the distribution of CMV gB genotypes in 14 patients with Posner–Schlossman syndrome and did not find significant differences in the clinical characteristics between patients infected with gB1 and those with gB3 [16]. The association between the different gB genotypes and patients’ clinical manifestations and outcomes has also been studied in nonophthalmic fields but with conflicting results. For example, gB3-related infection was identified as a risk indicator for CMV pneumonitis in hematopoietic stem cell transplant recipients. The gB3 genotype was reported to be correlated with a poor clinical prognosis in patients who had received bone marrow transplants [14]; gB3 and gB4 have also been reported to be associated with more invasive diseases [24,25]. However, other studies have not demonstrated a correlation between gB genotypes and outcomes [19,26,27]. These discrepancies could be attributed to differences in study populations, diseases, and patient immune status.

As mentioned, both a Japanese study and a Chinese study have identified gB1 or gB3 in patients with CMV anterior segment infection [15,16]; gB1 and gB3 were also predominant in our patients. Previous studies have indicated a high prevalence of gB2 in patients with AIDS with CMV retinitis, suggesting that different CMV genotypes may have different invasion pathways and tissue tropism [13,28]. Regarding the various manifestations of CMV anterior segment infection, we did not observe a significant difference in CMV genotype distribution between anterior uveitis and corneal endotheliitis, which is in agreement with the findings of the Japanese study [15]. These results indicate that CMV anterior uveitis and corneal endotheliitis are the same disease at different stages, which could be, at least partially, supported by the observation that patients with corneal endotheliitis were significantly older than those with anterior uveitis (64.0 ± 11.13 vs. 50.1 ± 12.27 years, *p* < 0.001). This signifies that the early phase of CMV anterior segment infection can manifest as anterior uveitis and that the later phase can manifest as endotheliitis. 

Our study has several limitations. First, our sample size was limited because of the relative rarity of the disease; nevertheless, our patient number is much larger than those in previous studies on gB genotypes in patients with CMV anterior segment infection. Second, we could not identify the individual titres of different gB genotypes in multiple-genotype infection because of the small number of aqueous specimens. Third, the current study was solely centred on the gB genotype, while the impact of other genotypes that could potentially influence clinical manifestation or outcome had not been evaluated [29,30,31]. A more detailed CMV strain characterization with more extensive sequencing strategies should be conducted in the future. Forth, various treatment protocols were applied to our patients. We should include more patients and stratify our population by treatment for more accurate results. Finally, inherent limitations in retrospective studies should also be taken into consideration when interpreting our study findings.

Although we did not identify a definite correlation between CMV gB genotypes and clinical characteristics in our patients, genotyping could still be applied to clarify some questions about CMV anterior segment infection in the future. First, when a patient who has received a CMV-positive corneal allograft acquires CMV infection, the possibility of graft-to-host transmission can be determined by exploring whether the virus genotypes in samples obtained from the patient’s aqueous humour differ from those in samples obtained from the donor cornea [32]. Second, for patients with recurrent CMV anterior segment infection, we can determine whether the recurrence is caused by the same strain as the previous infection. Genotype development was reported to be influenced by antiviral medication [33], which may explain why some of our patients had different CMV gB genotypes in different attacks. Longitudinal follow-up using aqueous humour analysis in these cases may help elucidate the relevance and pathophysiology between genotypes and medications. 

## 4. Materials and Methods

### 4.1. Study Population

We recruited all the patients with CMV anterior segment infection diagnosed at Chang Gung Memorial Hospital (CGMH), a tertiary referral centre in Taiwan, between 2015 and 2020. All patients manifesting characteristic signs of CMV anterior segment infection received a diagnostic aqueous tap. The aqueous humour was tapped with a 27-gauge needle during active inflammation, and the collected specimens were sent for quantitative PCR for CMV, herpes simplex virus (HSV), varicella-zoster virus (VZV), and EBV DNA detection. To extract viral DNA from the specimens, the High Pure Viral Nucleic Acid Kit (Roche Molecular Biochemicals, Mannheim, Germany) with a detection limit of 250 virus-derived DNA copies per millilitre for CMV DNA was used, as previously described [34]. 

CMV anterior segment infection was further classified into two categories: corneal endotheliitis and anterior uveitis. The diagnosis of CMV corneal endotheliitis was based on the criteria defined by the Japan Corneal Endotheliitis Study Group [35], which are (a) PCR (of aqueous humour samples) positive for CMV DNA but negative for HSV and VZV DNA and (b) clinical manifestations either for typical CMV endotheliitis: corneal endotheliitis with coin-shaped lesions or linear keratic precipitates (KPs); or for atypical CMV endotheliitis: corneal endotheliitis with localised corneal edema with KPs associated with two of the following: recurrent/chronic anterior uveitis, ocular hypertension/secondary glaucoma, and corneal endothelial cell loss. CMV anterior uveitis was diagnosed as evidence of anterior uveitis with PCR from aqueous positive for CMV DNA but negative for HSV and VZV DNA [34,36].

Medical records were reviewed for the demographics, medical history, and clinical findings of these patients, including age, sex, systemic diseases such as diabetes, hypertension, and other immune-related diseases, medication use, ocular surgery history prior to diagnosis, CMV viral loads, best-corrected visual acuity, IOP, and anterior chamber inflammation grade. Anterior chamber inflammation was observed using slit-lamp microscopy and graded according to standards set by the Standardization of Uveitis Nomenclature Working Group [37]. Corneal endothelial cells were evaluated through specular microscopy (CEM-530, Nidek Co., Ltd., Gamagori, Japan) at the center of the cornea in both the healthy and diseased eyes of the same patient. Once CMV anterior segment infection was confirmed, antiviral treatment was administered. Two treatment policies were applied: (a) intravitreal injection of ganciclovir (2 mg/0.05 mL) with or without oral valganciclovir depending on whether anterior chamber inflammation occurred after injection [38] or (b) prolonged use of topical 2% ganciclovir eye drops [39]. If an elevated IOP level with anterior chamber reaction or new onset of corneal edema was detected during the follow-up period, repeated aqueous tapping was performed to confirm the possibility of recurrence. The prognostic outcomes considered in this study included molecular recurrence within 1 year, the need for antiglaucoma medications, operation, or corneal decompensation. 

### 4.2. CMV gB Genotyping 

CMV gB genotyping was performed as previously described [17]. Briefly, we used extracted DNA as templates for the PCR amplification of specific regions of the CMV genome. For the first round of the CMV gB multiplex PCR assay, we used external forward and reverse primers. For the second round of the multiplex PCR assay, the PCR product was used. Five forward inner primers (designated as F1–F5) and one internal reverse primer were used for the second round of the multiplex PCR assay (Appendix A). The PCR product was visualised on 2% agarose gel. Appendix A lists the information on the primers. 

### 4.3. Statistical Analysis

In our study, descriptive analysis was used to summarise the characteristics of patients with CMV anterior segment infection. Between-group comparisons were performed using the Fisher exact test for categorical covariates and *t*-test for continuous covariates, respectively. We additionally calculated effect size, the standardized difference (STD), to express the size of the difference between the groups. An absolute value of STD greater than 0.2, 0.5, and 0.8 is considered to have small, medium, and large effect sizes, respectively [40]. For viral load and IOP levels, which were not normally distributed, between-group comparisons were performed using the Wilcoxon–Mann–Whitney two-sample rank-sum test. To evaluate the association between multiple genotypes and viral load (natural log-transformed), linear regression models were conducted with different covariates adjustment (e.g., endotheliitis; gB3 and endotheliitis). The linear regression model with adjustment of endotheliitis represented the analytic model of multiple genotypes and single genotype infection. Statistical significance was considered at a *p*-value of <0.05. Statistical analysis was performed using SPSS (IBM SPSS Statistics for Windows, Version 21.0.; IBM Corp., Armonk, NY, USA).

## 5. Conclusions

Our study indicated a predominance of gB1 and gB3 and a high prevalence of multiple gB genotypes in CMV anterior segment infection in the Taiwanese population. Multiple-genotype infection was correlated with relatively high CMV loads but not with specific clinical manifestations or prognostic outcomes. Furthermore, the gB3 genotype was correlated with a relatively high rate of filtering surgery before antiviral treatment, which may indicate poor IOP control. However, there was no difference among the outcomes in different genotypes. Additional studies enrolling more patients with a longer follow-up period should be conducted to elucidate the correlations and the pathophysiology of these results.

## Figures and Tables

**Figure 1 ijms-24-06304-f001:**
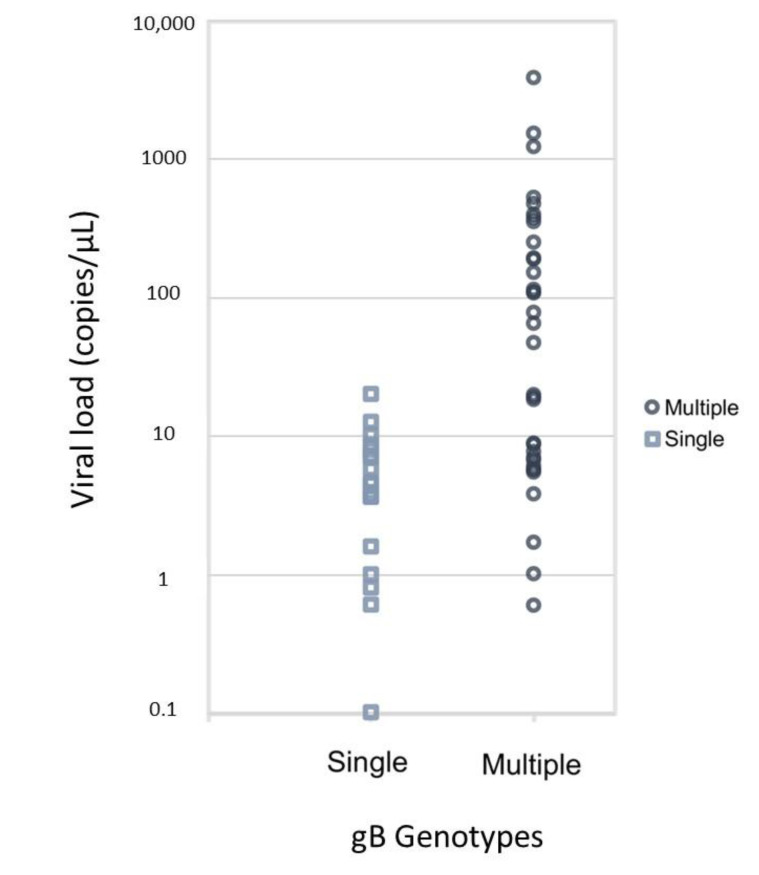
Single and multiple cytomegalovirus (CMV) glycoprotein (gB) genotype(s) and viral load, as measured by real-time PCR in aqueous sample from the patients with anterior segment ocular infection.

**Table 1 ijms-24-06304-t001:** Clinical characteristics and prognosis of patients with cytomegalovirus anterior segment infections with single and multiple genotypes.

	All (*n* = 57)	Infection with Single Genotype (*n* = 17)	Infection with Multiple Genotypes(*n* = 40)	STD	*p*-Value
General information					
Age (year)	58.6 ± 13.36	59.3 ± 13.45	58.4 ± 13.48	0.07	0.669
Past history					
Previous anti-viral use	7 (12.3%)	3 (17.6%)	6 (15%)	0.07	1
Previous glaucoma surgery	7 (12.3%)	3 (17.6%)	5 (12.5%)	0.14	0.684
Previous corneal transplant	6 (10.5%)	0	6 (15%)	−0.59	0.164
Clinical feature					
IOP (mm-Hg)					
Median [25th, 75th percentile]	22.9 [15.0–32.4]	23 [17–43]	23 [14–28]	-	0.339
Mean ± standard deviation	24.7 ± 12.2	28.3 ± 14.3	23.2 ± 11.1	0.40	0.149
Corneal edema	31 (54.4%)	10 (58.8%)	21 (52.5%)	0.13	0.780
KPs	49 (86.0%)	15 (88.2%)	34 (85%)	0.10	1
Diagnosis					0.234
Endotheliitis	35 (61.4%)	8 (47.1%)	27 (67.5%)	−0.42	
Iridocyclitis	22 (38.6%)	9 (52.9%)	13 (32.5%)	0.42	
Virology					
Viral load (copies/μL)					
Median [25th, 75th percentile]	8.6 [4.6, 112.1]	4.6 [1.6, 8.0]	56.0 [6.7, 252.0]	-	<0.001
Mean ± standard deviation	197.5 ± 592.4	5.3 ± 4.8	290.8 ± 706.4	−0.57	0.104
Mean ± standard deviation (log-transformed)	2.88 ± 2.29	1.1 ± 1.29	3.7 ± 2.21	−1.42	<0.001
Prognosis					
Recurrence in 1year (*n* = 46)	15(26.3%)	6 (35.3%)	9 (22.5%)	0.29	0.299
Anti-glaucomatic medication use	35 (61.4%)	13 (76.5%)	22 (55%)	0.46	0.207
Further filtering surgery	10 (17.5%)	4 (23.6%)	6 (15%)	0.21	0.471
Corneal decompensation	8 (14.0%)	3 (17.6%)	5 (12.5%)	0.14	0.701

STD, standardized difference; IOP: intraocular pressure; KP: keratin precipitates. Data are given as frequency (percentage) or mean ± standard deviation, unless specified.

**Table 2 ijms-24-06304-t002:** The distribution of cytomegalovirus glycoprotein B genotype in the patients with anterior segment infection.

	*n*	%
Single genotype	17	29.82%
gB1	4	
gB3	13	
Multiple genotype	40	70.28%
2 genotypes	21	36.84%
gB1 + gB3	5	
gB1 + gB4	10	
gB3 + gB4	6	
3 genotypes	19	33.33%
gB1 + gB2 + gB3	2	
gB1 + gB2 + gB4	4	
gB1 + gB3 + gB4	13	
Total	57	100%

gB: glycoprotein B.

**Table 3 ijms-24-06304-t003:** Comparison of clinical characteristics and prognosis between the patients with cytomegalovirus anterior segment infection with different glycoprotein genotype.

		gB1				gB3				gB4		
	gB1 (*n* = 38)	non-gB1 (*n* = 19)	STD	*p*	gB3 (*n* = 39)	non-gB3 (*n* = 18)	STD	*p*	gB4 (*n* = 33)	non-gB4 (*n* = 24)	STD	*p*
General information												
Age	58 ± 12.93	60 ± 14.45	−0.14	0.623	59.1 ± 14.35	57.7 ± 11.25	0.11	0.71	59.1 ± 13.23	58 ± 13.80	0.08	0.759
Past history												
Previous anti-viral use	5 (13.2%)	4 (21.1%)	−0.21	0.463	8 (20.5%)	1 (5.6%)	0.46	0.247	4 (12.1%)	5 (20.8%)	−0.24	0.470
Previous glaucoma surgery	4 (10.5%)	4 (21.1%)	−0.29	0.420	8 (20.5%)	0 (0%)	0.72	0.046	3 (9.1%)	5 (20.8%)	−0.33	0.261
Previous corneal transplant	6 (15.8%)	0 (0%)	0.61	0.164	3 (7.7%)	3 (16.7%)	−0.28	0.368	4 (12.1%)	2 (8.3%)	0.13	1
Clinical feature												
IOP (mm-Hg)												
Median [25th, 75th percentile]	25 [13, 30]	20 [15, 35]	-	0.876	20 [14, 35]	26 [17, 28]	-	0.317	21.5 [14, 26]	26 [16, 39]	-	0.145
Mean ± standard deviation	24.7 ± 12.7	24.8 ± 11.5	−0.01	0.961	23.8 ± 12.4	26.6 ± 12.1	−0.23	0.428	22.5 ± 10.2	27.7 ± 14.2	−0.42	0.113
Corneal edema	20 (52.6%)	11 (57.9%)	−0.11	1	21 (53.8%)	10 (55.6%)	−0.01	0.984	17 (51.5%)	14 (58.3%)	−0.14	0.488
KPs	33 (86.8%)	16 (84.2%)	0.07	1	33 (84.6%)	16 (88.9%)		1	28 (84.8%)	21 (87.5%)	−0.08	0.638
Diagnosis				0.155				0.579				0.413
Endotheliitis	26 (68.4%)	9 (47.3%)	0.44		23 (59%)	12 (66.7%)	−0.16		22 (66.7%)	13 (54.2%)	0.26	
Iridocyclitis	12 (31.6%)	10 (52.6%)	−0.44		16 (41.0%)	6 (33.3%)	0.16		11 (33.3%)	11 (45.8%)	−0.26	
Virology												
Viral load (copies/μL)												
Median [25th, 75th percentile]	19.1 [5.6, 192.4]	7.2 [3.6, 8.1]	-	0.039	8.6 [3.9, 78.3]	9.8 [4.6, 188.2]	-	0.519	65.3 [6.7, 350.1]	4.6 [1.6, 8.6]	-	<0.01
Mean ± standard deviation	294.3 ± 727.5	29.2 ± 79.6	0.51	0.121	79.4 ± 145.0	420.4 ± 965.2	−0.49	0.047	322.6 ± 745.8	12.7 ± 24.5	0.59	0.064
Mean ± standard deviation (log-transformed)	3.48 ± 2.39	1.85 ± 1.74	0.78	<0.01	2.59 ± 2.11	3.44 ± 2.58	−0.36	0.204	3.83 ± 2.24	1.48 ± 1.57	1.22	<0.01
Multiple-genotype infection	34 (89.5%)	6 (31.6%)	1.47	<0.01	26 (66.7%)	14 (77.8%)	−0.25	0.394	33 (100%)	7 (29.1%)	2.20	<0.01
Prognosis												
Recurrence in 1 year (*n* = 46)	9 (23.7%)	6 (31.6%)	−0.18	0.495	7 (17.9%)	4 (22.2%)	−0.11	0.566	7 (21.2%)	8 (33.3%)	−0.27	0.341
Antiglaucomatic medication use	22 (57.9%)	13 (68.4%)	−0.16	0.758	27 (69.2%)	8 (44.4%)	0.52	0.203	17 (51.5%)	18 (75%)	−0.50	0.076
Further filtering surgery	7 (18.4%)	3 (15.8%)	0.08	1	8 (20.5%)	2 (11.1%)	0.26	0.705	4 (12.1%)	6 (25%)	−0.34	0.298
Corneal decompensation	5 (13.2%)	3 (15.8%)	−0.03	1	7 (17.9%)	1 (5.6%)	0.39	0.415	3 (9.1%)	5 (25%)	−0.43	0.272

STD, standardized difference; IOP: intraocular pressure; KP: keratin precipitates.

**Table 4 ijms-24-06304-t004:** The association between viral load (log-transformed) and multiple genotypes in the linear regression model with different covariates.

Covariates Adjustment	*B* for Multiple Genotypes (95% CI)	*p* Value
None	2.57 (1.40, 3.74)	<0.001
Endotheliitis	2.55 (1.35, 3.76)	<0.001
gB1 and endotheliitis	2.40 (0.94, 3.87)	0.002
gB3 and endotheliitis	2.48 (1.26, 3.70)	<0.001
gB4 and endotheliitis	1.72 (−0.50, 3.94)	0.374

Abbreviations: *B*, regression coefficient; CI, confidence interval.

## Data Availability

Data are available upon request due to privacy or ethical restrictions.

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
