# Peer review of "Cytomegalovirus Glycoprotein B Genotype in Patients with Anterior Segment Infection"

_ijms, 2023, doi:10.3390/ijms24076304_

Round 1

Reviewer 1 Report

The study of Huang and collegues describes the investigation of the gB genotypes in aqueous humour samples of patients suffering from an CMV anterior segment infection. The sample size of the study cohort is large compared to other studies using similar patient cohorts. Also the kind of sample is very special and has rarely been investigated for CMV strain diversity. Hence, I believe it might be of substantial interest for the reader. However, gB genotyping was performed only by the use of gB genotype-specific primer pairs and visualization of the PCR products on the gel (as described by Tarrago et al., 2003). Although the authors described how the distinct gB genotypes are distributed and that mixed gB genotype infections are very frequent, no further sequence information is given.

Moreover, it would be interesting to learn whether the patients suffer from a CMV primary infection or reinfection/reactivation. Additionally, it would also be interesting to show if the patients have a CMV load also in other body compartments like blood, urine or saliva and if, whether the same gB genotypes can be found.

Collectively, the study is well done and concise. Unfortunately, a substantial limitation of this study is the simple gB genotyping strategy without determination of the sequence, the only focus on the gB cleavage site, no information on CMV in other body compartments, and no information on CMV primary or recurrent infection. In my opinion, a more detailed CMV strain characterization with more extensive sequencing strategies would increase the relevance of this study.

Minor comments

Due to the sensitivity limit of the PCR it is likely that multiple genotypes are predominantly found in samples with higher CMV load.

Major comments

Abstract

There is no background information given.

Line 17: 57 patients?

Introduction

It is not the latest knowledge which is given in the introduction. Relevant references are missing and some references are not appropriate. There is no clear explanation why gB cleavage site is used for genotyping. Hence, it is hard to understand why these findings may elucidate pathophysiology?

e.g. lines 38, 39, 44: references are missing

e.g. line 52; there is much more known about gB; not the appropriate reference

e.g. line 50; CMV genome encodes numerous glycoproteins; please specify

Author Response

Response to Reviewer 1 comments:

The study of Huang and collegues describes the investigation of the gB genotypes in aqueous humour samples of patients suffering from an CMV anterior segment infection. The sample size of the study cohort is large compared to other studies using similar patient cohorts. Also the kind of sample is very special and has rarely been investigated for CMV strain diversity. Hence, I believe it might be of substantial interest for the reader. However, gB genotyping was performed only by the use of gB genotype-specific primer pairs and visualization of the PCR products on the gel (as described by Tarrago et al., 2003). Although the authors described how the distinct gB genotypes are distributed and that mixed gB genotype infections are very frequent, no further sequence information is given.

Moreover, it would be interesting to learn whether the patients suffer from a CMV primary infection or reinfection/reactivation. Additionally, it would also be interesting to show if the patients have a CMV load also in other body compartments like blood, urine or saliva and if, whether the same gB genotypes can be found.

Collectively, the study is well done and concise. Unfortunately, a substantial limitation of this study is the simple gB genotyping strategy without determination of the sequence, the only focus on the gB cleavage site, no information on CMV in other body compartments, and no information on CMV primary or recurrent infection. In my opinion, a more detailed CMV strain characterization with more extensive sequencing strategies would increase the relevance of this study.

Response: Thank you very much for your constructive feedback on this manuscript. Your insightful comments have provided us with several directions for future studies. We appreciate your suggestions and will collect more samples from patients to conduct the study accordingly. We have also described some points in the limitations of the manuscript. “The current study was solely centered on the gB genotype, while the impact of other genotypes that could potentially influence clinical manifestation or outcome had not been evaluated. A more detailed CMV strain characterization with more extensive sequencing strategies should be conducted in the future.” (page 8, lines 228)

Minor comments

Due to the sensitivity limit of the PCR it is likely that multiple genotypes are predominantly found in samples with higher CMV load.

Response: Thank you very much for your nice comment. It could be one of possible explanations for the phenomenon. On page 7, line 179, we have added it as “Due to the sensitivity limit of the PCR, it is likely that multiple genotypes are predominantly found in samples with higher CMV load.”

Major comments

Abstract

There is no background information given.

Response: Thank you for your comment. We have added the background as “The glycoprotein B (gB) on viral envelope, encoded by most widely characterized polymorphic gene gpUL55, is responsible for cytomegalovirus entry into host and could serve as a potential marker of pathogenicity.” in the abstract. 

Line 17: 57 patients?

Response: Sorry for the typo. We have corrected it.

Introduction

It is not the latest knowledge which is given in the introduction. Relevant references are missing and some references are not appropriate. There is no clear explanation why gB cleavage site is used for genotyping. Hence, it is hard to understand why these findings may elucidate pathophysiology?

Response: Thank you very much for your comment. We have changed the introduction as “The human CMV genome encodes numerous glycoproteins, including glycoprotein B(gB), glycoprotein H(gH), glycoprotein L(gL), and glycoprotein O(gO). Glycoprotein B found on the CMV envelope, encoded by the UL55 gene, has been extensively studied, and is known to play a critical role in various viral activities such as viral entry into hosts, cell-to-cell viral transmission, and fusion of infected cells [1]. CMV gB gene has the potential to serve as a potent marker of pathogenicity and a viable target for treatment;vaccine targeting gB was found to reduce post-transplantation CMV viremia in phase II clinical trial [2]. ” (page 2, lines 50-56)

e.g. lines 38, 39, 44: references are missing

Response: Thanks for your reminder. We have added the following two references..

  1. “Forte E, Zhang Z, Thorp EB, Hummel M. Cytomegalovirus Latency and Reactivation: An Intricate Interplay With the Host Immune Response. Front Cell Infect Microbiol. 2020;10:130.”
  2. “De Venecia G, Zu Rhein GM, Pratt MV, Kisken W. Cytomegalic inclusion retinitis in an adult. Arch Ophthalmol. 1971;86(1):44-57.”

e.g. line 52; there is much more known about gB; not the appropriate reference.

Response: As we know, gB5 was discovered few years after the identification of gB1-gB4 [3]. Previous clinical research seldomly discussed gB5 as it was rarely found in serum samples[4]. In fact, we did include the inner primer F5 in our multiplex PCR assay, but we did not detect any gB5 in our samples. Thank you very much for your reminder.  We have updated our manuscript to include the mention of gB5 in the introduction section as “There are five primary gB genotypes, namely gB1, gB2, gB3, gB4, and gB5 identified so far” (page 2, line 57), method section as “” (page 10, line 291), result section as “No gB5 was detected in the sample.” (page 3, line 88), supplementary table (primer of gB5), and added the references accordingly. Please allow me to attach the supplementary table here.

e.g. line 50; CMV genome encodes numerous glycoproteins; please specify.

Response: Thank you for the suggestion. We have specified the glycoproteins on CMV on page 2, line 51. “The human CMV genome encodes numerous glycoproteins, including glycoprotein B (gB), glycoprotein H (gH), glycoprotein L (gL), and glycoprotein O (gO).”

References:

  1. Navarro D, Paz P, Tugizov S, Topp K, La Vail J, Pereira L(1993) Glycoprotein B of human cytomegalovirus promotes virion penetration into cells, transmission of infection from cell to cell, and fusion of infected cells.Virology, 197(1):143-158.
  2. Griffiths PD, Stanton A, McCarrell E, Smith C, Osman M, Harber M, Davenport A, Jones G, Wheeler DC, O'Beirne J et al(2011) Cytomegalovirus glycoprotein-B vaccine with MF59 adjuvant in transplant recipients: a phase 2 randomised placebo-controlled trial.Lancet, 377(9773):1256-1263.
  3. Shepp DH, Match ME, Lipson SM, Pergolizzi RG(1998) A fifth human cytomegalovirus glycoprotein B genotype.Res Virol, 149(2):109-114.
  4. Tarrago D, Quereda C, Tenorio A(2003) Different cytomegalovirus glycoprotein B genotype distribution in serum and cerebrospinal fluid specimens determined by a novel multiplex nested PCR.J Clin Microbiol, 41(7):2872-2877.

Reviewer 2 Report

In this retrospective study, the authors determined the HCMV glycoprotein B genotypes in 57 patients with anterior segment eye infection in Taiwan.  They analyzed the correlation between diggerent genotypes and clinical characteristics, based on the assumption that HCMV gB genotypes might be correlated to virulence and pathogenicity in anterior segment infection. The authors concluded that eye infections with multiple genotypes are common in Taiwan. Genotypes gB1 and gB3 were the most frequent ones. Multiple-genotype infection correlated with higher viral loads, but did not correlate with specific clinical manifestations or prognostic markers.

The strength of the study is the relatively large number of patients, considering that this type of infection is rare. The study also appears to fit well with the topic of the special issue. Weaknesses include the rather descriptive nature of the study, the focus on a single glycoprotein gene for genotyping, and the ultimately negative outcome, i.e. the inability to correlate the genotype to a clinical presentation or disease severity. However, the authors openly discuss the limitations of the study.

Specific comments:

1.       How was the sample size (number of patients) determined? Were all available patients included, or were inclusion and exclusion criteria defined? Could there be a selection bias (admission rate bias, non-respondent bias, etc.)

2.       This disease mainly occurs by reactivation of latent CMV infections. Therefore, information on chronic diseases that affect the immune status of the enrolled patients should be included.

3.       Why did the authors focus on gB? Other more polymorphic genes (e.g., gO) have been used for genotyping. Did the authors consider other genes?

4.       The genotype can be influenced by antiviral treatment, particularly in mixed infections. Can the authors show data regarding the genotype(s) pre and post antiviral treatment?

5.       P values do not inform the reader about the strength or magnitude of the effect. Including additional statistical parameters would strengthen the study.

6.       There are fifteen patients experienced recurrence within 1 year, is there any genotype changes in the follow-up?

Reviewer 3 Report

In their current manuscript, Huang et al. present an interesting study on the prevalence of single or multiple genotype CMV infections in patients with ocular infections. The study design is appropriate and the manuscript is very well-written, I only have a few comments that the authors should address:

The abstract does not do the manuscript justice, please modify especially the background section, which should not only be a statement of the aim of the study.

Lines 53-56: results of these other studies should be briefly mentioned

line 88: "past history" could mean a lot of things, please specify

table 3: please explain how the log transformation was done, I don't see how 19.1 transforms to 3.48 (for gB1)? Does the log-transformed viral load give the value per ml? I'm assuming you are showing mean and range, and mean +/- SD for the viral loads? Please clarify in the revised manuscript

discussion line 161-162: you mention a high prevalence of CMV infection in Taiwan, can you provide numbers and is it possible to compare to other countries, specifically those where other gB genotype studies were performed?

line 248: typo "endothiliitis"

Round 2

Reviewer 1 Report

Dear authors,

thank you very much for the careful revision of your manuscript. It would be of high interest to learn more about the HCMV strains in these specific patient samples which you mentioned is planned for future studies.

Reviewer 2 Report

The authors have responded adequately to my comments.